# Fast and Simple Protocol for N-Glycome Analysis of Human Blood Plasma Proteome

**DOI:** 10.3390/biom14121551

**Published:** 2024-12-04

**Authors:** Denis E. Maslov, Anna N. Timoshchuk, Alexander A. Bondar, Maxim P. Golubev, Anna G. Soplenkova, Maja Hanic, Sodbo Z. Sharapov, Olga N. Leonova, Yurii S. Aulchenko, Tatiana S. Golubeva

**Affiliations:** 1MSU Institute for Artificial Intelligence, Lomonosov Moscow State University, Moscow 119192, Russia; bochlit2@gmail.com (D.E.M.); timoschukanny@gmail.com (A.N.T.); emaxya@yandex.ru (M.P.G.); anna.mail10@mail.ru (A.G.S.);; 2Institute of Chemical Biology and Fundamental Medicine SB RAS, Novosibirsk 630090, Russia; alex.bondar@mail.ru; 3Genos Glycoscience Research Laboratory, 10000 Zagreb, Croatia; mhanic@genos.hr; 4Priorov Central Institute of Traumatology and Orthopedic, Moscow 127299, Russia; 5Institute of Cytology and Genetics SB RAS, Novosibirsk 630090, Russia; 6Immanuel Kant Baltic Federal University, Kaliningrad 236041, Russia

**Keywords:** N-glycosylation, N-glycans, xCGE-LIF analysis, human blood plasma, post-translational modifications, APTS labeling

## Abstract

N-glycome analysis of individual proteins and tissues is crucial for fundamental and applied biomedical research and medical diagnosis and plays an important role in the evaluation of the quality of biopharmaceutical and biotechnological products. The interest in this research area continues to grow annually, thereby increasing the demand for the high-throughput profiling of human blood plasma N-glycome. In response to this need, we have developed an optimized, simple, and rapid protocol for the N-glycome profiling of human plasma proteins. This protocol encompasses the entire analysis cycle, from plasma isolation to N-glycan spectrum quantification. While the proposed method may have lower efficiency compared to already published high-throughput methods, its adaptability makes it suitable for implementation in virtually any molecular biological laboratory.

## 1. Introduction

Protein glycosylation is one of the most common post- and co-translational modifications that significantly enhances the diversity of proteins [1,2]. The conformation of proteins, along with their properties and activity, is considerably influenced by the attachment of oligosaccharides [3]. This affects receptor affinity, substrate specificity, and protein half-life [1]. Glycosylation primarily occurs in the Golgi apparatus, with a minor presence in the endoplasmic reticulum. The unique characteristics of glycosylation enzymes enable the production of numerous oligosaccharide combinations, resulting in vast diversity within the human glycoproteome [4,5]. The regulation of this diversity is influenced by genetic, epigenetic, and environmental factors [6].

The glycosylation profile typically remains stable within certain boundaries in healthy individuals [7]. However, it can experience significant alterations due to pathological conditions such as type I and type II diabetes [8], malignant tumors [9,10], cardiovascular diseases [11,12], Parkinson’s disease [13,14], and other disorders [15]. Additionally, the glycan profile is known to be associated with a person’s gender, age, and lifestyle [16,17]. An example of age-related changes in immunoglobulin G (IgG)-linked glycan structures includes an increase in bisection and a decrease in galactosylation and sialylation [16]. In contrast, during pregnancy, the opposite pattern is observed, with increased galactosylation and sialylation and decreased bisection [18].

Human saliva, serum, and plasma are easily accessible biological materials that can be used for N-glycome analysis [19]. N-glycome profiling can provide significant insights into health and disease. For example, a recent report has highlighted the possibility of using N-glycan analysis in conjunction with genotyping to identify Alzheimer’s disease at its early stages [20,21,22].

Currently, the process of obtaining a glycan profile is considered to be relatively simple and routine. However, it does require specialized and expensive equipment, thus restricting the capacity to conduct such studies without substantial funding. We propose a newly developed protocol that is based on the same biochemical principles as the methodology for obtaining N-glycan fractions from human blood plasma described by Reiding et al. [23] and Hennig et al. [24]. We succeeded in adapting the process for more accessible equipment available in almost any molecular biology laboratory (amplifier, shaker, centrifuge, capillary sequencer). Our method has somewhat lower throughput but does make the analysis of N-glycosylation more accessible.

## 2. Materials and Methods

### 2.1. Materials

The analysis was conducted using blood samples provided by four participants of a disease-oriented Russian disc degeneration study (RuDDs). Plasma sampling was performed at the Novosibirsk Research Institute of Traumatology and Orthopedics according to a previously published protocol [25].

The following reagents were used: SDS, Igepal CA-630, PBS, APTS (8-Aminopyrene-1,3,6-Trisulfonic Acid, Trisodium Salt), 2-picoline borane, DMSO, citric acid, acetonitrile and triethylamine (Sigma Aldrich, St. Louis, MO, USA), Biogel P10 (Bio-Rad, Hercules, CA, USA), LIZ500 length marker and Hi-Di Formamide (Applied Biosystems, Carlsbad, CA, USA), and PNGase F (Promega, Madison, WI, USA).

The laboratory plasticware included 1.5 mL tubes and 0.2 mL PCR tubes (Axygen Scientific, Inc., Union City, CA, USA), a 0.22 µm filter mini column with 2 mL tubes with lids (Nordic Biosite, Täby, Sweden), and hematology tubes with EDTA (ApexLab, Moscow, Russia).

The following equipment was used: +4 °C refrigerator with temperature re-mode, −80 °C Kelvinator, CM-6MT centrifuge (Elmi, Riga, Latvia), 5418R centrifuge (Eppendorf, Nijmegen, the Netherlands), OS-20 orbital shaker (Biosan, Riga, Latvia), vortex V-1 plus (Biosan, Riga, Latvia), M111-02-96 amplifier (Bis-N, Koltsovo, Russia), and 3130XL automated gene analyzer (Applied Biosystems, Thermo Scientific, Waltham, MA, USA).

The data were processed and analyzed using custom in-house Python scripts available upon reasonable request. Electrophoretic signal processing and subsequent demarcation of migration peaks were performed using the SciPy [26] and NumPy [27] libraries. Dynamic time warping (DTW) was performed using the DTAIDistance python module [28].

### 2.2. Methods

#### 2.2.1. Blood Plasma Preparation

All blood samples in the study were obtained from healthy participants. The collection of blood samples from participants was performed using hematology tubes. The tubes were subjected to a gentle mixing process to prevent bubbling. Following this, they were incubated at room temperature for a period of 30 min. Subsequently, centrifugation was performed at a speed of 1100 rcf for 10 min at room temperature to facilitate plasma separation. The next step involved transferring the plasma into 1 mL tubes and placing them in a Kelvinator operating at −80 °C for preservation. Prior to the isolation of N-glycans, the sample underwent thawing, followed by the extraction of a plasma aliquot for the analysis and subsequent freezing of the remaining sample.

#### 2.2.2. Isolation of N-Glycans

A total volume of 4 μL of a 2% SDS solution dissolved in water was introduced into a PCR tube containing 2 μL of human plasma. The contents were mixed thoroughly using a pipette. Subsequently, the tubes containing the samples were put into an amplifier, with the reaction mixture being maintained at a temperature of 65 °C and the lid temperature set at 104 °C for 10 min. Next, an aliquot of 2 μL from an aqueous solution containing 8% Igepal CA-630 was added and mixed meticulously through pipetting. The resulting mixture was left on an orbital shaker at 450 rpm for 3 min to induce protein denaturation.

While the mixture was being incubated on the shaker, the necessary steps were taken to prepare the PNGase F solution for glycosylation: 2 μL of 5×PBS with 0.12 μL of PNGase F per sample were mixed. A volume of 2 μL from this solution was introduced into the blood plasma sample. The mixture was thoroughly agitated, sealed, and incubated within the amplifier for 3 h (at a reaction mixture temperature of 37 °C and lid temperature of 104 °C). During this stage, plasma proteins were subjected to deglycosylation.

Once the deglycosylation process was complete, a volume of 2 µL of the reaction mixture was transferred into a new PCR tube. Following that, a mixture of 2 μL APTS (30 mM APTS in 3.6 M citric acid) and 2 μL of 2-picoline borane (1.2 M in DMSO) was carefully added to the tube by gently pouring it along the inner wall. Once sealed, the tube was vortexed for 15 s at the lowest speed and then placed in an amplifier for 16 h. The reaction occurred at a temperature of 37 °C, while the lid temperature was maintained at 104 °C. This procedure resulted in fluorescently labeled N-glycans.

The reaction was halted by introducing 100 μL of cold 80% acetonitrile in water, and the tubes were then refrigerated until the columns were prepared for glycan purification (incubation at a maximum of 37 °C for 1 h). The column preparation involved the use of 200 μL of solvent, which was a 10% BioGel P10 solution made up of water, acetonitrile, and 96% ethanol in a ratio of 7:2:1. This solvent was applied to a membrane filter column and then centrifuged at 1000 rcf for 1 min to eliminate the solvent. Next, 200 μL of water was added to each column, and the columns were centrifuged once more at 1000 rcf for 1 min. The step of rinsing the columns with water was performed two more times, bringing the total number of water rinses to three. Then, 200 μL of an 80% acetonitrile solution in water was added to the column, followed by centrifugation at 1000 rcf for 1 min to eliminate the solvent. Thorough cleaning was ensured by washing the columns with 80% acetonitrile three times, with two additional repetitions of this step. The completion of these procedures made the columns suitable for purifying glycans.

The reaction mixture containing APTS-labeled N-glycans and excess dye was vigorously mixed using a vortex, following which the entire volume of the mixture (106 μL) was carefully transferred to the prepared column. The column was then placed on an orbital shaker, operating at a speed of 100 rpm, for 5 min. To remove the solvent from the column, a centrifugation step was used, which lasted for 1 min at a speed of 200 rcf.

Next, to remove excess dye, the column was washed with 200 μL of an 80% acetonitrile solution in water, which contained 100 mM trimethylamine. The column was placed on an orbital shaker (100 rpm) for 2 min and then centrifuged to remove the solvent (1 min, 200 rcf). The part of the procedure from the addition of 200 µL of 80% acetonitrile in water solution containing 100 mM trimethylamine until centrifugation was repeated 4 more times. As a result, the column should be subjected to 5 washes with this solution.

Afterwards, the N-glycan sample was washed to ensure the complete removal of trimethylamine. For this purpose, a 200 μL aliquot of an 80% acetonitrile solution in water was introduced into the column. The column was subsequently positioned on an orbital shaker set at 100 rpm for a duration of 2 min, followed by centrifugation at 200 rcf for 1 min to eliminate the solvent. Following the addition of 200 µL of an 80% acetonitrile solution in water, the centrifugation step should be repeated twice. Additionally, the column must be washed a total of three times with this solution.

Elution was carried out by applying 50 μL of water to the column, followed by placing the column on an orbital shaker set at 100 rpm for 5 min. Following that, the column underwent centrifugation at a speed of 200 rcf for a duration of 2 min, resulting in the collection of the eluent in a sterile tube. Next, 100 μL of water was applied to the column. The column was placed on an orbital shaker (100 rpm) for 5 min and then centrifuged (2 min, 200 rcf). The eluent was collected into a tube with the previous fraction. The process was subsequently repeated once more. The total volume of solution containing the N-glycan fraction in the tube should amount to approximately 200 µL.

#### 2.2.3. Separation of the N-Glycan Fraction by Capillary Gel Electrophoresis

A working mixture consisting of 3 µL of N-glycan fraction, 1 µL of LIZ500 size standard diluted to 1:50, and 6 µL of HiDi formamide (total volume of 10 µL) was prepared for analysis. The separation of fluorescently labeled N-glycans was performed on a 3130XL genetic analyzer calibrated using the G5 dye set in an 80 cm long 16-capillary array filled with NimaPOP-7 polymer (NimaGene). A standard electrophoresis module (GeneScan 80 POP7) for fragment analysis on the 80 cm capillary assembly was modified as follows. The oven temperature parameter was decreased to 30 °C. Injection voltage and time were increased to 15 kV and 20 s, respectively. After the injection of the samples prepared in HiDi formamide, capillary electrophoresis was conducted at a standard voltage of 14.6 kV and a modified oven temperature of 30 °C. Data were collected over 7000 s and then analyzed.

#### 2.2.4. Electropherogram Analysis

The data generated by the sequencer in ABIF format were used to extract “DATA 1” and “DATA 105”, corresponding to a wavelength with maxima at 522 nm and 655 nm for APTS-labeled N-glycome specter and LIZ-labeled oligonucleotide comigrated standard, respectively. Subsequently, migration scale normalization was carried out using the migration data of the oligonucleotide standard, and the target region was precisely excised within a range of 150 to 350 nucleotides. The signal was denoized using a one-dimensional Gaussian filter [29] with a standard deviation equal to 5 measurement points. Subsequently, baseline correction was carried out utilizing the ARPLS method [30]. The determination of peak maxima involved identifying local maxima in the processed spectra, with a height and topological prominence of no less than 1% of the highest point in the glycomic region. The peak boundaries were determined by taking into account the standard deviation of the Gaussian curve fitted to the peak, with a range of ±3 standard deviations from the center point of the maximum. Subsequently, the area of the peak was calculated using Simpson’s integration method.

Detected peaks were annotated by comparison to a reference spectrum from Reiding et al. [23]. The annotation process involved aligning the spectra using DTW. A match between a pair of peaks was established if an optimal warping path connected these two peaks. Additionally, manual peak matching was performed through visual inspection. Structures corresponding to matched peaks in Reiding et al.’s [23] reference spectrum were then assigned to the corresponding peaks detected in this study.

## 3. Results

In order to assess the efficacy of the method proposed, blood samples were collected from four donors. For every sample, blood plasma and an N-glycan fraction were extracted. Each sample underwent a total of four technical repetitions.

The electropherograms obtained through fractionation on the sequencer exhibited a signal indicative of human plasma N-glycome. This signal aligns with the reference one [23], both in terms of the electrophoretic migration pattern within the glycomic region (located between the 150 and 350 nucleotide markers) and the peak magnitudes (Figure 1A,B). It is worth noting that the glycomic region in our study starts ~20 nucleotides earlier than that in the reference one. The observed differences in the migration pattern of the LIZ-labeled size ladder relative to the glycan signal between our study and the reference study can be attributed to the reduced field strength employed in our experiments (14.6 kV/80 cm) versus that used in the reference study (15 kV/50 cm). While the mobility of smaller analytes is generally independent of electric field strength, long polymer chains, such as DNA fragments, exhibit altered electrophoretic mobility under higher field strengths due to the biased reptation model [29]. Overall, we identified 30 distinct peaks with heights of at least 1% of the maximum peak height in the glycomic region of at least one electropherogram. Of these, 28 peaks were annotated as corresponding to the structures from Reiding et al. [23] (Figure 1C, Table 1). One peak (Peak 0 in Figure 1C) did not match the reference electropherogram, and another peak (Peak 4) resulted from the overlapping of the tails of two adjacent peaks and therefore was excluded from the downstream analysis.

In order to assess the reproducibility of the method, we analyzed the synchrony of the resulting electrophoretic migration patterns and the variability of peak heights and peak areas for individual peaks on four biological samples of human plasma N-glycome obtained through four technical repetitions. The electropherograms were characterized by significant visual resemblance (Figure 2), as well as substantial numerical measures of synchrony: the average Pearson correlation coefficient for the spectra of the glycomic regions was 0.897, and the average median lag of the maximum cross-correlation was 4 measurement points (see Table 2). The variability in peak areas and maxima is similar to that observed in the reference method [23]. Within single biological samples, we calculated the median of the coefficient of variation for maximum height-normalized peak heights and total area-normalized peak areas. For both metrics, the median coefficient of variation did not exceed 0.05. A notable difference can be observed when comparing this with the coefficient of variation for peak areas in the reference study (Figure 3), which is 0.0992. It is important to highlight that the present study exclusively focuses on major peaks within the glycomic region, with heights amounting to at least 1% of the maximum peak height in at least one electropherogram. Notably, the total count of these peaks (29; Peak 4 was not included in the analysis due to it originating from the overlap of the adjacent peak tails) is lower in comparison to the reference study (49). The augmented variability observed in the latter could potentially be attributed to the inclusion of additional and more noisy minor peaks. Therefore, we can confidently state that N-glycans can be effectively extracted from human blood plasma and analyzed using our approach.

## 4. Discussion

We have introduced an effective approach for characterizing human plasma N-glycome, comprising blood sample collection, plasma isolation, and subsequent N-glycan fractionation. The protocol involves the denaturation of glycoproteins in an SDS solution, followed by the liberation of the N-glycan fraction using the PNGase F enzyme. Subsequently, APTS dye is added to label the glycans with fluorescence. The unbound dye is removed through the process of gel filtration. The samples obtained are fractionated and subsequently detected on an ABI 3130xl DNA Analyzer, with the addition of a DNA standard. The main advantage of our method, when compared to the method described in [23], is the availability of the equipment employed. Specifically, the mini centrifuge designed for plates was substituted with a vortex designed for PCR tubes; the thermostat designed for PCR plates with a temperature gradient was substituted with a PCR amplifier; the 96-well membrane plates intended for sample washing were replaced with membrane columns; and a centrifuge was employed instead of a plate vacuumator. This method allows one to isolate up to 48 N-glycan samples per day and measure up to 384 samples in one run of the instrument. Despite the throughput being slightly inferior to similar techniques (because of the use of tubes instead of 96-well plates in the glycan isolation step), it grants more researchers the ability to study N-glycosylation due to its more straightforward instrumentation.

The primary output of this technique is an N-glycome fingerprint, which can be used to analyze the similarities of different biological samples, both in bulk and by individual peaks. By supplying an N-glycome fingerprint with annotations for individual peaks—obtained either through cross-referencing with existing migration pattern databases for N-glycan species or by conduction glycosidase sequencing—researchers can uncover valuable insights into fundamental biological processes, linking alterations in N-glycome profiles to human health and disease. Moreover, the developed protocol can be extended to other N-glycome fractions, enhancing its utility. Specifically, the steps of deglycosylation, fluorescent labeling, and subsequent fractionation by capillary gel electrophoresis enable the generation of distinct electrophoretic migration patterns for individual N-glycoproteins, such as immunoglobulin G (IgG) and transferrin.

## 5. Conclusions

The cost-efficiency, combined with the analytical potential, of the developed technique enables more laboratories to conduct frequent N-glycome analyses across various biological contexts and human conditions. The ability to link N-glycome changes to potential biomarkers can greatly enhance our understanding of disease mechanisms and contribute to the development of targeted therapies. Although our sample preparation method resulted in lower yields, it is beneficial in that it allows N-glycome analysis at a reasonable cost.

## Figures and Tables

**Figure 1 biomolecules-14-01551-f001:**
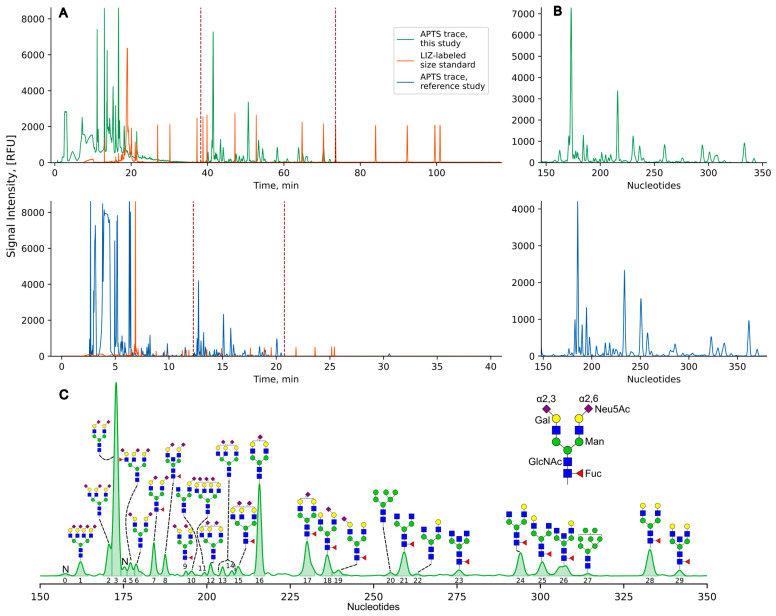
The electropherograms of human plasma protein N-glycome samples generated using the capillary gel electrophoresis with laser-induced fluorescence (CGE-LIF) method. (**A**) Electrophoretic migration patterns for N-glycome samples generated in this study (upper plot) and in the reference study (lower plot) [23]. The *X*-axis shows the electrophoretic migration in minutes. Green and blue denote APTS-labeled N-glycan traces, and orange denotes LIZ-labeled co-migrating size standard trace. The target signal boundaries are indicated by dashed lines. In the reference study, electrophoresis was performed using 50 cm capillaries at 15 kV for 40 min. (**B**) The zoom of the target signal region. The *X*-axis shows electrophoretic migration in nucleotide coordinates. (**C**) Peaks detected in this study. The numbers under the spectrum are identifiers for the peaks. N denotes the absence of annotated structures. Some signals correspond to multiple glycan structures. For the full assignments of each peak detected, see Table 1.

**Figure 2 biomolecules-14-01551-f002:**
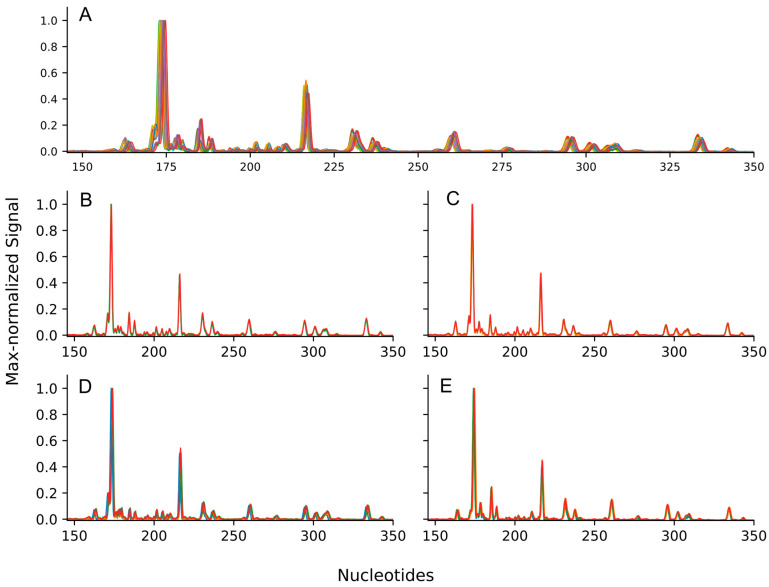
The synchronicities of N-glycome profiles of human plasma proteins generated using CGE-LIF. (**A**) Overlapping electropherogram of 16 technical replicates from four biological samples, showing the combined migration pattern over time. (**B**–**E**) Individual overlaid electropherograms for technical replicates of samples 1 to 4, respectively. The *X*-axis represents the migration time in nucleotide coordinates, while the *Y*-axis represents the relative signal intensity.

**Figure 3 biomolecules-14-01551-f003:**
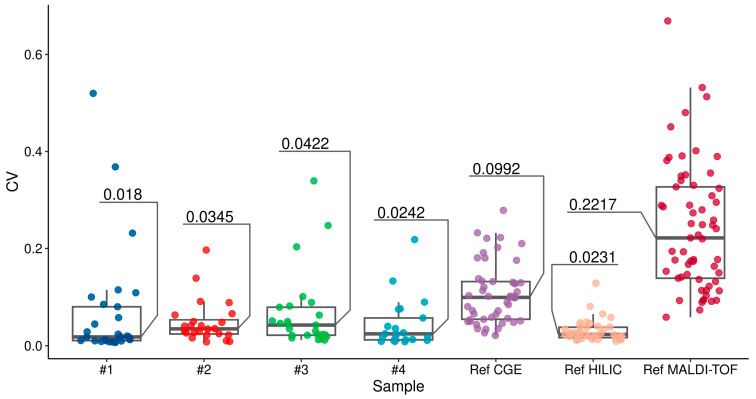
The reproducibility of the method developed and the HILIC, CGE, and MALDI-TOF methods on the reference data [23]. This figure illustrates the estimated coefficient of variation for total-area normalized peak areas, along with an indication of the median value in one sample. Sample is the in-study sample identifier.

**Table 1 biomolecules-14-01551-t001:** Annotation of the detected peaks.

Peak ID	DTW	Visual	Result	Structure	Peak ID	DTW	Visual	Result	Structure
0	NA	NA	NA ^a^	NA	15	19	19	19, 20 ^d^	A3F1G3S2 A3F1G3S[6,6]2 FA2G1S[6]1 FA3G3S[6,6]2
1	1	1	1	A4G4S[6,6,6,6]4	16	22	22	22	A2G2S[6]1 FA2[6]BG1S[6]1
2	3	3	3	A3G3S[3,6,6]3	17	25	25	25	FA2G2S[6]1 M5
3	4	4	4	A2G2S[6,6]2	18	26	26	26	FA2BG2S[6]1 FA2[6]G2S[3]1
4	5	NA	NA ^b^	NA	19	27	27	27	FA2[3]G2S[3]1
5	5, 6	NA	5 ^b^	A3F1G3S[3,6,6]3	20	30	30	30	M6 A3G3S[6]1
6	6	6	6	A2G2S[3,6]2	21	31	31	31	FA2 A3G3S[6]1
7	7	7	7	FA2G2S[6,6]2	22	32	32	32	A2[6]G1
8	9	9	9	A1[3]G1S[3]1 FA2G2S[3,6]2	23	35	35	35	FA2B
9	11	11	11	FA2G2S[3,3]2 A4G4S[3,3,6]3	24	39	39	39	FA2[6]G1
10	15	NA	12 ^c^	FA1[3]G1S[6]1 FA1[6]G1S[3]1 A2[6]G1S[6]1	25	40	40	40	FA2[3]G1
11	15	15	15	A4G4S[3,3,3]3	26	41	41	41	A2G2 FA2[6]BG1
12	16	16	16	A3G3S [6,6]2	27	41	42	42 ^e^	M8 FA2[3]BG1
13	17	17	17	A3G3S[3,6]2	28	45	45	45	FA2G2 M9
14	18	18	18	A3G3S[3,6]2	29	46	46	46	FA2BG2

Peak ID entifies peaks detected in this study (Figure 1C). DTW is the corresponding peak ID from Reiding et al. [23], based on DTW alignment. Visual is the peak ID from Reiding et al. that visually corresponds to the peak in this study. Result is the peak ID from Reiding et al., chosen as the corresponding match(es) for the peak in this study. Structure is the structural annotation in Oxford notation from Reiding et al. A: antennary N-acetylglucosamine; B: bisecting N-acetylglucosamine; G: galactose; M: mannose; F: fucose; S: N-acetylneuraminic acid. Numbers in brackets indicate the the branching pattern of the glycan. a: Likely an artifact peak. b: Peak 4 likely resulted from the overlap between the tails of Peaks 3 and 5. In contrast, Peak 5 shows higher variability across biological samples, which cannot be attributed to the effect of the overlap. c: Not prominent in the reference spectrum and corresponds to Peak 12 in Hennig et al.’s work [24], annotated with the same structure as Peak 12 in Reiding et al.’s work. d: Peaks 19 and 20 together form a plateau in the spectrum, which are not visually separated in Figure 1 for Reiding et al.’s work. These two overlap with Peak 15 in this study; therefore, Peak 15 is assigned to A3F1G3S2 as a major structure. e: Peak 27 is the trailing peak in a distinct series of four peaks and is visually distinct from the preceding three.

**Table 2 biomolecules-14-01551-t002:** Statistics for signal reproducibility and synchronization.

Sample	Pearson’s ρ	Pearson’s ρ, MH-Normalized	Median Lag of MCC	Median CV, TA-Normalized Peak Area	Median CV, MH-Normalized Peak Height
No. 1	0.985	0.883	2.50	0.018	0.0147
No. 2	0.97	0.758	1.50	0.0345	0.0407
No. 3	0.795	0.616	9.50	0.0422	0.0371
No. 4	0.836	0.728	2.50	0.0242	0.0241

Sample is the in-study sample identifier. Pearson’s ρ is the average Pearson correlation coefficient for the raw spectra. Pearson’s ρ, MH-normalized is the average Pearson’s correlation coefficient for the maximum height-normalized spectra. Median lag of MCC is the median lag of the maximum cross-correlation. Median CV, TA-normalized peak area is the median coefficient of variation for the total area-normalized peak areas. Median CV, MH-normalized peak height is the median coefficient of variation for the maximum height-normalized peak heights.

## Data Availability

Data are contained within the communication.

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
