# Peer review of "Fast and Simple Protocol for N-Glycome Analysis of Human Blood Plasma Proteome"

_biomolecules, 2024, doi:10.3390/biom14121551_

Round 1
Reviewer 1 Report
Comments and Suggestions for Authors
The authors of this article have focused on the limitations of N glycan analysis in the small-scale laboratory and have developed a protocol for N-glycan analysis, that can be implemented in any molecular biology laboratory, without the need of specialized and expensive equipment. The authors used human blood plasma and developed an optimized, simple, and rapid protocol for N-glycome profiling. The authors also found that their proposed method, even though is slightly inferior inn throughput compared to similar techniques, can grant more researcher the ability to study N-Glycosylation. I think that the authors have done a good job of developing this protocol, however, to make this article strong, further validation is required. Unfortunately, I cannot recommend this article for publication in this journal.
I think the authors should include more samples to further validate their methods. It would be a good idea to include some samples, where the authors can compare a normal vs diseased state to see if this method can give more insights on the differences in the glycan composition of normal vs diseased samples.
Materials and methods:
Does the blood sample from the four participants is disease oriented. If yes, did author include any normal blood/serum samples?
Line 99: What was the PNGase F enzyme concentration for the deglycosylation?
In Figure 1: The LIZ-label size standard looks different when compared to the reference.
Also, Positions of the peaks (APTS trace) are also slightly different compared to the reference study. Can they include explanations, about the differences?
For figure 2: I would suggest authors to include the color legends for each sample.
The attached supplementary information is not in English. If authors provide any supplementary information, it should be in English, so that its easier for review.
Reviewer 2 Report
Comments and Suggestions for Authors
Comments
In this study, Maslov et al. established a new protocol for the preparation of N-glycan samples from human plasma for the N-glycome analysis using capillary gel electrophoresis with laser-induced fluorescence detection via 8-aminopyrene-1,3,6-trisulfonic acid (APTS) labeling. Although this sample preparation method resulted in lower yields, it is beneficial in that it allows N-glycome analysis at a reasonable cost.
This study contains some of novel findings and the results seem sound. However, some revisions should be required in regards to the following points.
Major concerns:
If this study was performed based on the previous reference study by Reiding KR (2019) (Ref. 12), all peaks (29) should be annotated to the predicted N-glycan structures in the reference. Please indicate the N-glycan structures for the peaks detected in Fig. 1B or Fig. 2.
Minor concerns:
Page 2, line 64: Igepal means IGEPAL CA630?
Page 2, line 64: Full chemical name should be provided for APTS.
Page 4, line 181: The word “electrophoregram” must be “electropherogram”.
Page 5, line 206 (Fig. 1 legend): The word “electrophoregram” must be “electropherogram”.
Page 5, line 207 (Fig. 1 legend): Full name should be provided for CGE-LIF.
Reviewer 3 Report
Comments and Suggestions for Authors
In the manuscript „Fast and simple protocol for N-glycome analysis of human blood plasma proteome” the authors presented their modification of capillary gel electrophoresis based method for the analysis of N-glycans.
The work is based on the method of Reiding et al. (2019). The authors described their modification in detail, and as the main advantage pointed the broad availability of equipment used, making the glycan analysis more available to the research community. This method can be regarded as high throughput because it allows us to isolate 48 samples daily and to analyze up to 384 samples in one run of the instruments. In addition, the authors stated that the reproducibility of their methods is very high, with coefficients of variation for repeated measurement of the same sample lower than 0.05.
I think that this article can be published in Biomolecules, after minor modification.
Minor modification:
Line 54 – References 12 and 13 are nor cited appropriately. Reference 12 is by Reiding and colleagues and Reference 13 is by Hennig and colleagues.
Line 99 – The authors should write exact concentration of PNGase F .
Line 108 – The authors should give full name of ATPS reagent, not only the abbreviation.
Round 2
Reviewer 2 Report
Comments and Suggestions for Authors
I am pleased to say that the manuscript was much improved in its revised form.